# Smartphone-Based Methodology Applied to Electromagnetic Field Exposure Assessment

**DOI:** 10.3390/s24113561

**Published:** 2024-05-31

**Authors:** Pablo-Luis López-Espí, Rocío Sánchez-Montero, Jorge Guillén-Pina, Rubén Castro-Sanz, Ricardo Chocano-del-Cerro, Juan-Antonio Martínez-Rojas

**Affiliations:** Department of Signal Theory and Communications, University of Alcala, 28801 Alcalá de Henares, Spain; rocio.sanchez@uah.es (R.S.-M.); jorge.guillenp@uah.es (J.G.-P.); ruben.castros@edu.uah.es (R.C.-S.); ricardo.chocano@uah.es (R.C.-d.-C.); juanan.martinez@uah.es (J.-A.M.-R.)

**Keywords:** EMF exposure, electromagnetic pollution, risk assessment, smartphone, kriging

## Abstract

This study presents the measurements of exposure to electromagnetic fields, carried out comparatively following standard methods from fixed sites using a broadband meter and using a smartphone on which an App designed for this purpose has been installed. The results of two measurement campaigns carried out on the campus of the University of Alcalá over an area of 1.9 km^2^ are presented. To characterize the exposure, 20 fixed points were measured in the first case and 860 points along the route made with a bicycle in the last case. The results obtained indicate that there is proportionality between the two methods, making it possible to use the smartphone for comparative measurements. The presented methodology makes it possible to characterize the exposure in the area under study in four times less time than that required with the traditional methodology.

## 1. Introduction

Concern about exposure to radio frequency (RF) electromagnetic fields (EMF) is an issue of growing interest in modern society. The evolution of the network towards 5G technology has increased this concern. In the words of the European Commission, “Some citizens perceive the fifth generation of wireless networks—5G—as a threat to public health, as they think that EMF exposure is higher than exposure from current 4G networks” [1]. The spread of alarming news about the issue and the skepticism of the population can only be combated with dissemination tools that contribute to an adequate perception of the risk. The clear presentation of the results in the form of exposure maps is one way to contribute to an adequate perception of the exposure levels and their temporal evolution. Interpolation techniques such as kriging are used to produce these exposure maps [2,3]. In addition, this type of large-scale study allows for further analysis related to the appearance of or increase in different diseases. 

According to commonly accepted procedures, there are two types of measurements for monitoring RF field exposure: mobile (personal) and fixed. Fixed measurements, determined with a spectrum analyzer or a frequency-selective meter, are the most accurate, although they require more effort in terms of cost and personnel [4]. For rapid characterization at a fixed location, there is also the possibility of using a broadband probe. When the location to be assessed is very large, it is of crucial importance to optimize the number of measurements to be performed to be able to carry them out without increasing the cost of the equipment or greatly increasing the number of hours required. Mobile (personal) measurements use exposure meters that allow many measurements to be collected for a single individual or a population group. They allow for the analysis of spatial variation for a single user but present the problem of generalization of the obtained results. When measurements of multiple environments are required, the number of exposure meters needed or the time needed to carry out the study is also increased if the measurements are taken consecutively. The cost of the equipment (thousands of Euros per device) is a very important limiting factor for both fixed measurements and personal exposure meters. This has led to the search for low-cost solutions for the measurement of the new 5G mobile networks [5].

Up to now, to ensure the adequacy of EMF exposure levels at a site regarding the current regulations, two methods are recommended: measurements with a selective meter or spectrum analyzer and broadband measurements averaged over a 6 min period. Both are based on the original 1998 ICNIRP guidelines [6], EN50413 and IEC 62311 [7], on which the European regulations are based [8]. There has recently been an update of the ICNIRP recommendations on EMF protection [9]. Among the most notable changes affecting this work are the change in the averaging interval to 30 min (as opposed to the previous 6 min) for the whole body and the limit of 2 GHz in the definition of the reference values. These changes have not yet been incorporated into the legislation. For this reason, and to facilitate comparison with previous work, the measurements made in this proposal were averaged over a 6 min interval. 

The novelties of this study are as follows:A smartphone-based method is proposed to obtain a large number of measurements of an area in which to characterize EMF exposure.An area of 1.9 km^2^ was established by installing the terminal on a bicycle. Exposure maps were made using the equivalent interpolation techniques for fixed sites. The obtained results were compared with those of standard methodologies.A tool is provided for obtaining values equivalent to EMF exposure that can, in a simple way, to contribute to an adequate risk perception.

The proposal is organized as follows. Section 2 describes the location of the area under study and the measurement points for the cases of fixed sites and the smartphone. It also includes the characteristics of both devices and briefly mentions the programming tools used to design the application that allows for the cell phone to be used as a power meter. The relationship between both measured values is also theoretically justified. Section 3 discusses the measurements taken with both methods, their statistical distribution, and the possibility of generating exposure maps with both methods. Section 4 discusses the advantages and limitations of the presented proposal. Finally, Section 5 highlights the conclusions of this study.

## 2. Materials and Methods

### 2.1. Location of the Measurement Area

The University campus of the University of Alcalá is located on the outskirts of the town of Alcalá de Henares (Spain). It is an area of approximately 1.9 km^2^ that contains several faculties, a university hospital, and an industrial and technology area. There are six mobile telephone base stations (BTS) on or near the campus. The location of the area and the base stations is shown in Figure 1.

In line with previous work [2], the surface of the campus was divided into grids of 250 m on each side and those grids whose surface was mostly in the line of sight (LOS) with one of the existing BTS were located. On the surface, 47 grids were identified, in which 20 locations were determined for the realization of fixed measurements. The density of points reached was 10.52 points/km^2^, with an average distance of 300 m between them. Of the total number of measurements, 9 corresponded to LOS situations and 11 to NLOS. Figure 2 shows the grids and measurement points that were identified. A total time of 4 h was used to carry out these 20 measurements during 5–8 May 2024.

Following this procedure, the surface of the campus was routed along the trajectories shown in Figure 3. A total of 860 points were taken over a period of 1 h and 10 min on 9 May 2024.

### 2.2. Measurement Equipment and Protocol

For the measurement of the electric field intensity at fixed sites, a Narda EMR-300 Broadband RF Survey Meter and a Narda Isotropic Probe 18C from Narda Safety Test Solutions GmbH (Pfullingen, Germany) were used in the 100 kHz to 3 GHz range with 0.01 V/m resolution, a detection level of 0.2 V/m, a dynamic range of 60 dB, a linearity of ±0.5 dB, an isotropic deviation of ±1.0 dB, an rms measurement range of 0.2 to 320 V/m, and a sensitivity 0.2 V/m. We also utilized a non-metallic Tripod EMCO 11689C (Figure 4, left). The equipment was placed in a clear area within the chosen grids, and the electric field measurement was performed using the built-in averaging function during a six-minute interval. To perform the measurements with the smartphone, a holder was installed on a bicycle (Figure 4, right) and a route was planned through all the streets of the university campus. A period of 5 s was selected for taking measurements, although the application designed for the terminal allows the interval to be varied from 1 s. The smartphone used was a OnePlus Nord N10 5G.

Although it is not the final objective of this proposal, to adequately detail the used methodology, a brief description of the development methodology followed for the design of the app used to take measurements with a smartphone is presented as follows. For the design of the App, the Flutter environment for open-source software development was used. This environment, which uses the Dart programming language, was created by Google for building high-quality user interfaces for mobile, web, and desktop from a single code base [10,11]. Visual Studio Code was used as the main code editor to develop the application; this is an editor that supports Flutter and Dart, allowing for the code to be debugged [12]. Finally, to compile the application for Android devices, Android Studio was chosen [13]. The objective of this application was to obtain the power values received from all the BTS that were accessible from the device. To do this, the application must obtain the READ_PHONE_STATE permission. This will grant access to the “ConnectivityManager”, which contains the device connectivity information. Subsequently, the App will receive the “TelephonyManager”, which specifically focuses on managing and accessing information related to mobile telephony and the type of mobile network to which the device is connected. Finally, it iterates through all the cells to which the device has access (using the function “getAllCellInfo”) to obtain the signal strength in dBm and the network type of each cell. The obtained cells may vary between devices due to device capacity or provider restrictions. Each measure is dated and geo-referenced using the terminal’s own time and GPS information and finally sent to a parse database for final storage [14]. As indicated above, the objective of the application was focused on obtaining a value proportional to the EMF exposure due to cell phone signals, but it would be possible to determine the rest of the wireless services detected by the cell phone (Bluetooth, WiFi, etc.). Further details of the application or its code can be obtained by contacting the authors. This also implies that these signals, which can be acquired by the EMR300, are not included in the measurements taken with the smartphone, which could be a source of deviation between the values taken by both methods.

### 2.3. Relation between Smartphone and Broadband Meter Measurements

The electric field values obtained by the Narda EMR300 meter were taken in V/m, while the signal levels indicated by the smartphone correspond to the power (in milliwatts or its equivalent in decibels, dBm) captured by the mobile phone antenna. The relationship between the electric field and the acquired power can be established following some radio propagation calculations. The distance of the measurement points (in the case of both the Narda meter and the smartphone) to the BTS can be used to assume the far-field condition and to make the plane wave assumption. In this case, it is possible to simplify the calculation of the relationship between the incident electric field and the power detected by the terminal. The power density module (S) associated with an electromagnetic wave, as a function of its electric field value, is given by Equation (1):(1)$$S=\frac{{\left|E\right|}^{2}}{120\pi }$$

The relationship between the power acquired by an antenna (the value indicated by the smartphone) and the incident power density is given by the effective area (S_eff_), given by Equation (2), where G is the isotropic gain of the smartphone antenna and λ is the wavelength of the RF signal:(2)$$S_{eff}=\frac{\lambda^{2}}{4\pi }G$$

Combining the above expressions, the measured electric field can be related to the power captured by the mobile. If logarithms are also taken to express the electric field in dBV/m and the power in dBm, the following expression is obtained:(3)$$P\left(\mathrm{dBm}\right)=E\left(\mathrm{dBV}/m\right)+20log\left(\lambda \left(m\right)\right)+G\left(\mathrm{dB}\right)+66.76\;\mathrm{dB}$$

Therefore, for a given frequency and a specific gain, the relation in logarithmic units between power and electric field is an offset value. 

Since the value of the mobile antenna gain depends on the orientation of the phone with respect to the different BTS, the above expression is not easy to calculate. However, this result allows us to infer that the representation of the exposure maps, using the logarithmic measurements in both methods, should differ in an offset factor and, therefore, the spatial variability of both representations should be similar. Having proved this, it would be possible to carry out the measurement campaigns using a mobile terminal, with the consequent economic savings, if the offset between the two could be subsequently determined. This equivalence between measurement methods was similarly established when comparing the results obtained using personal exposure meters and broadband meters in [15,16], using a proportionality value between the two methods.

## 3. Results

First, the statistical distribution of both data sets was analyzed. On the one hand, the distribution of the values measured with the EMR 300 broadband meter was fitted to the distribution, and on the other hand, the values measured with the smartphone were fitted to the distribution. Since the values measured with the app are given in dBm, to compare the statistical distribution, the equivalent electric field was obtained. Since no realistic gain estimate is available, it was decided in this case to normalize the maximum value to unity. In both cases, in accordance with previous work, the values were fit to a lognormal distribution. Statistical analyses were carried out using Statgraphics 19 [17] software. This package can determine the means and variances, compare sample independence, and fit points to different probability density functions. The results of both adjustments can be seen in Figure 5 and Figure 6.

As can be seen, both measurement sets follow the lognormal distribution fit. Clearly, the statistical properties of the mean and variance of both sets are not the same, but this is perfectly explainable by the exact conversion value between both measurement methods being unknown. Additionally, the measurement condition itself is different in both cases. In the case of the meter, it is a measurement averaged over a 6 min interval at the same location; in the case of the smartphone, it is a power measurement received by the terminal taken every 5 s along the indicated route. Measurements taken with the smartphone are subject to greater variability (fading or loss of signal) and may therefore lead to problems in the collected data. A further statistical analysis in Section 3 discusses the statistical validity of measurements taken along a trajectory instead of in a fixed point.

Therefore, a rigorous calibration process would be necessary to establish a direct relationship between the two measurements. This is also useful in terms of the existence of proportionality between measurements. This result could lead to an immediate application for the establishment of comparisons, for example, between indoor and outdoor exposure, using a mobile terminal, or variations in exposure between different areas. Following this idea, EMF exposure maps were obtained using both techniques. Figure 7 shows the interpolation obtained using the values measured with the EMR300 meter. Figure 8 shows the interpolation obtained using the values taken with the smartphone. The maps were created using ArcGIS Pro 3.0 [18]. The methodology chosen for the geographical interpolation of exposure values was ordinary kriging.

The maps show the BTS in the study’s area of influence with red dots. The fixed locations where measurements were taken with the EMR300 are indicated by red dots (LOS) and red squares (NLOS) on the map in Figure 7. The points measured along the trajectory taken with the bicycle are indicated by lines on the map in Figure 8.

The results shown in Figure 7 and Figure 8 prove that both methods can detect the areas of greatest exposure in an equivalent manner. The greatest differences are seen in the northern and southeastern zones. This is an artifact of the interpolation due to the lack of sampling points in the perimeter of the area. An island effect is also seen in some of the areas of the exposure map generated from the broadband meter values. This effect is possibly due to insufficient sampling in some areas. The average distance between measurement points was 300 m, which is at the limit obtained in previous work [2]. A larger range of values also appeared in the case of measurements taken with the smartphone (approximately 40 dB vs. 20 dB). That is, in the case of the broadband meter, a range of values from the maximum to a 100 times lower value is obtained, while in the case of the smartphone, a range of values from the maximum to a 10,000 times lower value is obtained. This is due to the different sensitivities of both meters.

Table 1 shows the measured values for the fixed sites and the interpolated values at the same locations taken from the map in Figure 8. The mean difference between the measured and interpolated EMF values is 72.3 dB for LOS cases and 67.8 dB for NLOS cases. This result is also consistent with the one obtained in [16], where two conversion constants were also observed between exposure meters and broadband meters depending on the LOS/NLOS situation.

To investigate the relation between the results of the different methods, measurements were taken at three additional points, marked A, B, and C in Figure 2. At each point, a measurement was taken with the EMR300 meter averaged over 6 min. Simultaneously, smartphone measurements were taken every second. One smartphone was fixed on the same measuring tripod as the EMR300 meter and the other measurements were taken while walking around the fixed point (see the yellow dots on the inner map in Figure 2). 

The density traces of the measurements made with the fixed smartphone and with the itinerant one are shown in Figure 9. 

Measurements taken while moving around the same point have lower values and greater variability than those taken at a fixed point. This may be due to multiple factors: other wireless services present, the reception of several frequencies simultaneously, different directions of arrival, different sensitivities of the meter and the smartphone, temporary signal fades, etc. But, as can be seen in Figure 9, the statistical distribution of the signals received in the six-minute interval is similar in both cases. This leads to the conclusion that small movements around a site follow a similar trend to single-point measurements. In other words, a continuous measurement at a site can be replaced, without noticeable statistical difference, by a trajectory around the site. Therefore, when cycling a route, nearby points behave statistically as a single fixed point. 

A different question is the possibility of establishing a direct relationship between the two measurements. Table 1 compares the results of the interpolation with the kriging of both methods, reaching differences in the range of 60 to 87 dB between the values measured with the EMR300 and the values measured with the smartphone. Table 2 shows the results corresponding to the three additional points that were analyzed. Points A and B present LOS conditions with respect to the nearest emitter, while point C corresponds to an NLOS condition. 

Table 2 shows that there is a relationship between the measured values in the sense that a higher exposure value measured with the EMR300 corresponds to a higher value of power detected by the smartphone. Beyond that, with the available data, it seems difficult to expect that a smartphone can easily replace an exposure meter.

However, from the point of view of the characterization of large surfaces, its value has been proven as an element that can efficiently detect the areas of greatest exposure and, therefore, its greatest usefulness will be in the determination of sensitive areas where later measurements can be made with calibrated instrumentation.

## 4. Discussion

In this proposal, we analyzed the possibility of using a smartphone to carry out measurements equivalent to those of EMF exposure and thus contribute to an adequate risk perception. The proposed system presents, firstly, advantages in terms of the cost of the equipment needed to obtain the measured values. This could allow for collaborative studies or studies with a large number of volunteers, making it possible to generalize the results, which is the main limitation of the methods based on exposure meters. It also allows many measurements to be obtained over time, similar to what happens with personal exposure meters, although this point was not the subject of this work. The existence of an offset factor (in dB) between the exposure values measured with standard methods and the values collected with the smartphone makes this method suitable for comparative studies.

A methodology for exposure mapping was also proposed, in this case, by cycling through an area. The methodology, combined with interpolation techniques, proved to be equivalent to that based on traditional measurement methods. It also offers the advantage of less time being invested in data collection. For the 1.9 km^2^ area under study, the time taken to complete the route with the bicycle was 1 h and 10 min. In the case of the fixed measurement sites, each of them was averaged for 6 min. We must add the time corresponding to the travel between points, so the total time spent in this method was 4 h. This significant reduction in the required time provides an advantage.

The main drawback of the proposed methodology is that, lacking a precise conversion factor, the values do not faithfully represent the exposure, but a proportional value. Another limitation of the proposed smartphone measurement technique is that it is focused only on the measurement of BTS signals. Although the measurement possibilities can be extended to other wireless technologies, such as WiFi or similar technologies, contributions to exposure due to TV or FM are excluded from the measurement possibilities.

In terms of risk perception, the contribution can be significant since it provides the public with a comparative and transparent tool of the received values. The application allows the user to learn, in real time, the values of the set of signals (of the different BTS) to which he is exposed and to establish his own comparisons of the different exposure situations in which he may find himself. In addition, the agile creation of exposure maps using the measured values contributes to the effective and accessible dissemination of the exposure of the different areas. Furthermore, these values can be contrasted with those measured by the user himself, helping to increase confidence in the published data.

## 5. Conclusions

In this study, we compared the performance of EMF exposure measurements and maps following the measurement methodologies for fixed points with a broadband meter and sampling from the data collected by a smartphone. To calculate the sum of the powers received by the cell phone, an application was designed that uses the possibilities of Android terminals.

The data collected by the terminal are proportional to the exposure measurements and follow similar statistical and spatial distributions, so they are useful for comparative studies and provide the advantages of automation and less time being spent in gathering them.

The accessibility of the received signal data in any terminal in which the application is installed helps to generate confidence in the exposure values by making it possible to contrast the collected values with those that can be obtained by any user of the application.

In this proposal, we presented an alternative for the rapid characterization of an area with respect to cell phone emissions. It is not intended to replace the measurement by means of a personal exposure meter with a smartphone. For this, an extension of the research work to a much more exhaustive characterization of the terminal would be necessary to establish a relationship between the exposure values and the instantaneous values. The complete characterization of the gain of the cell phone is also indispensable to determine this relationship. Similarly, the influence on the exposure of services not covered in this work (FM, DTV, WiFi, etc.) must be assessed when comparing the two values.

## Figures and Tables

**Figure 1 sensors-24-03561-f001:**
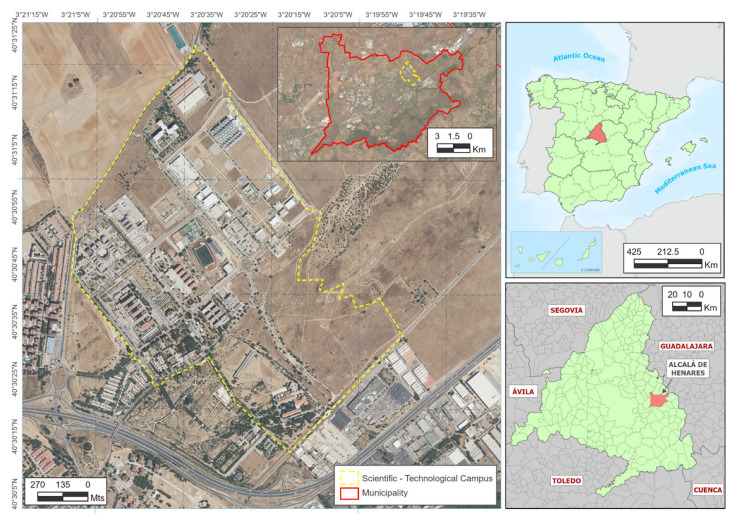
Location of measurement area.

**Figure 2 sensors-24-03561-f002:**
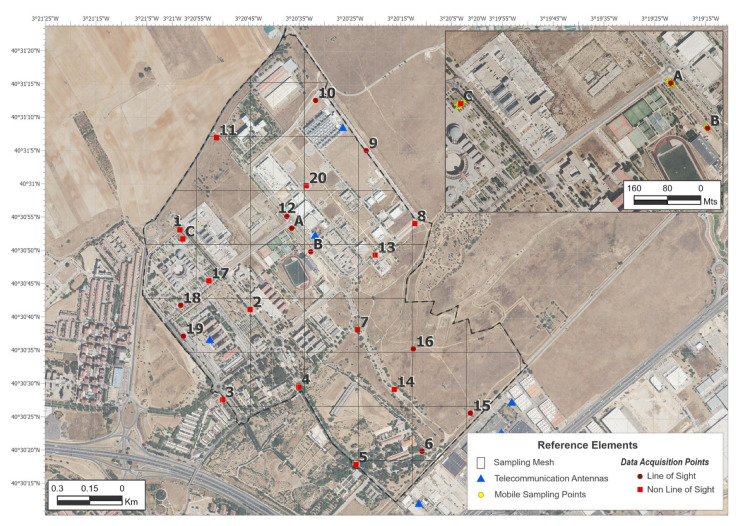
Measurement grid and fixed points location. Inner map of statistical analysis points.

**Figure 3 sensors-24-03561-f003:**
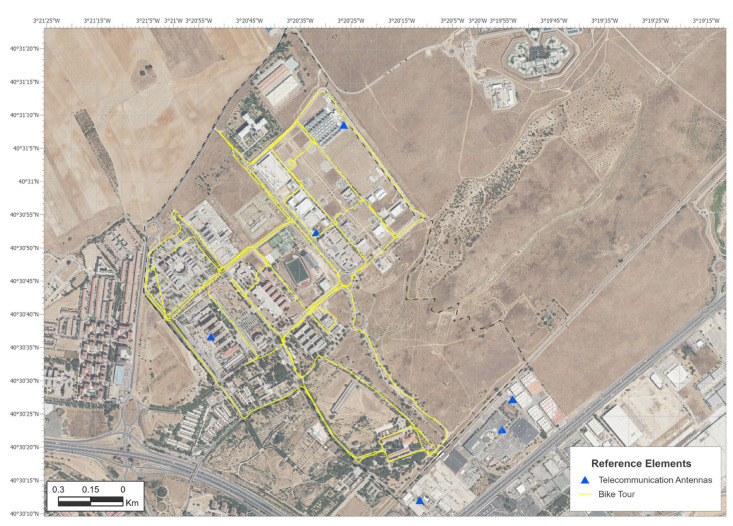
Bicycle tour path location.

**Figure 4 sensors-24-03561-f004:**
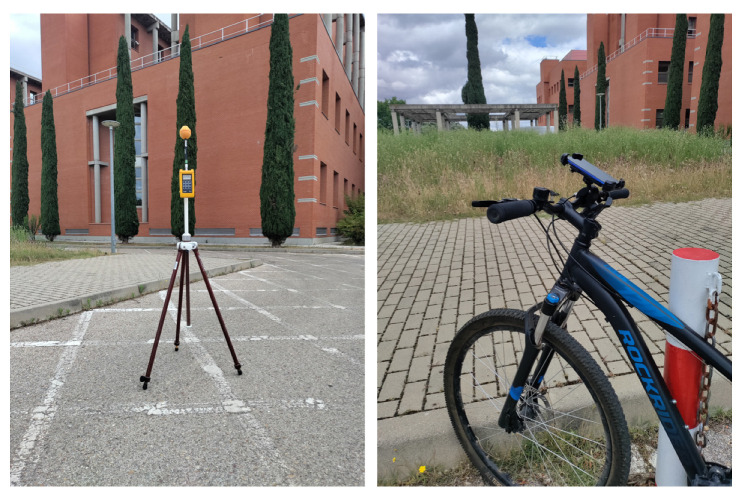
Broadband Narda EMR300 meter (**left**) and bicycle used for the tour with the mobile terminal placed in its holder (**right**).

**Figure 5 sensors-24-03561-f005:**
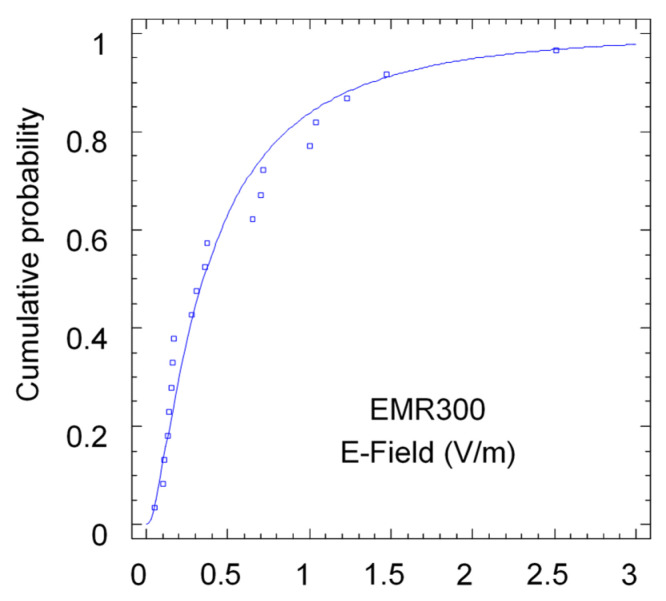
Cumulative probability of electric field values measured with EMR300 meter.

**Figure 6 sensors-24-03561-f006:**
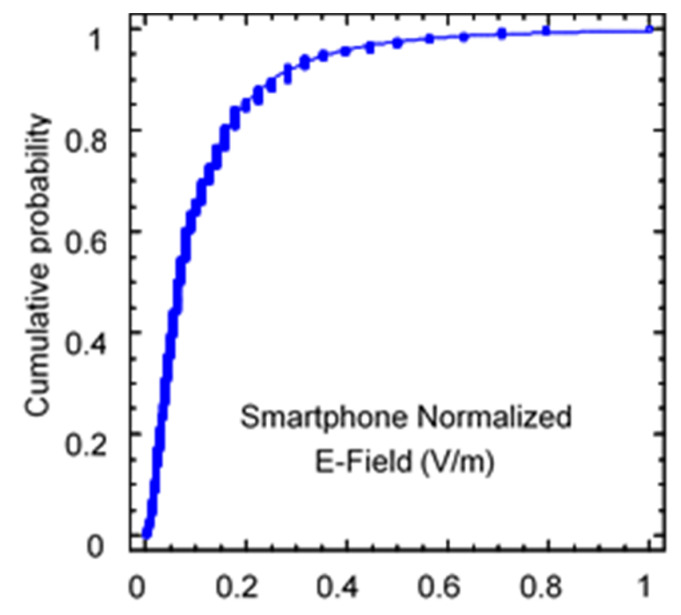
Cumulative probability of normalized field values measured with a smartphone.

**Figure 7 sensors-24-03561-f007:**
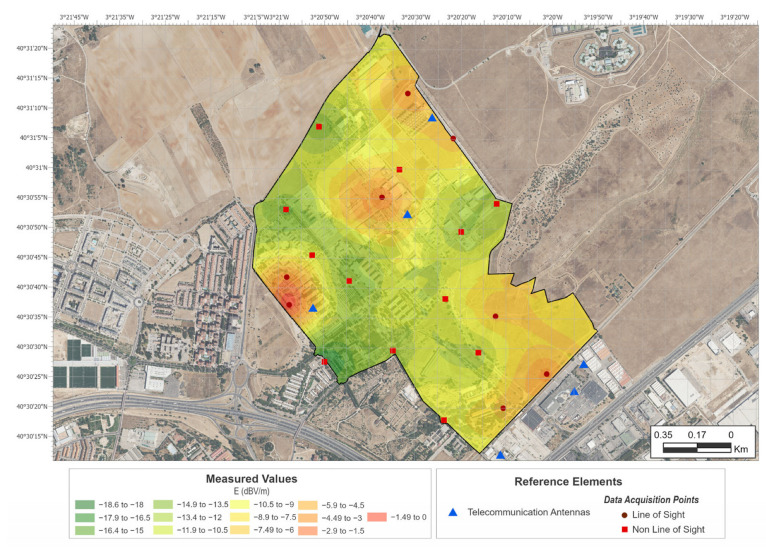
EMF exposure map obtained from fixed points’ kriging interpolation.

**Figure 8 sensors-24-03561-f008:**
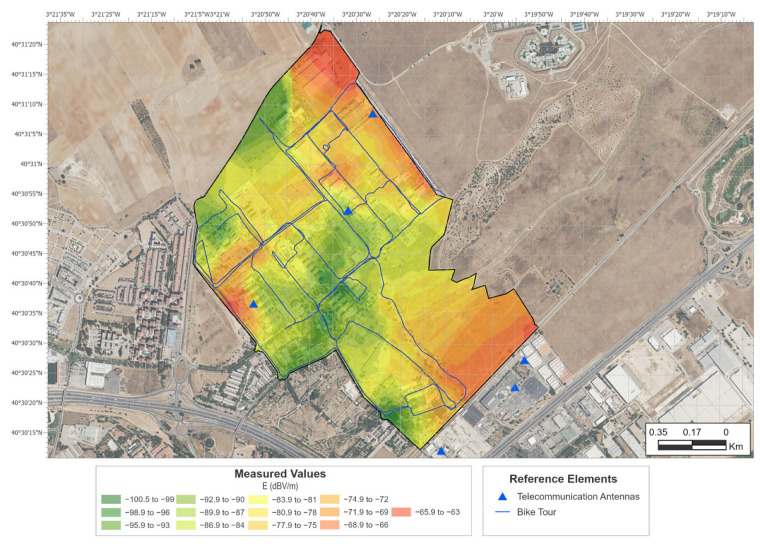
EMF exposure map obtained from smartphone points’ kriging interpolation.

**Figure 9 sensors-24-03561-f009:**
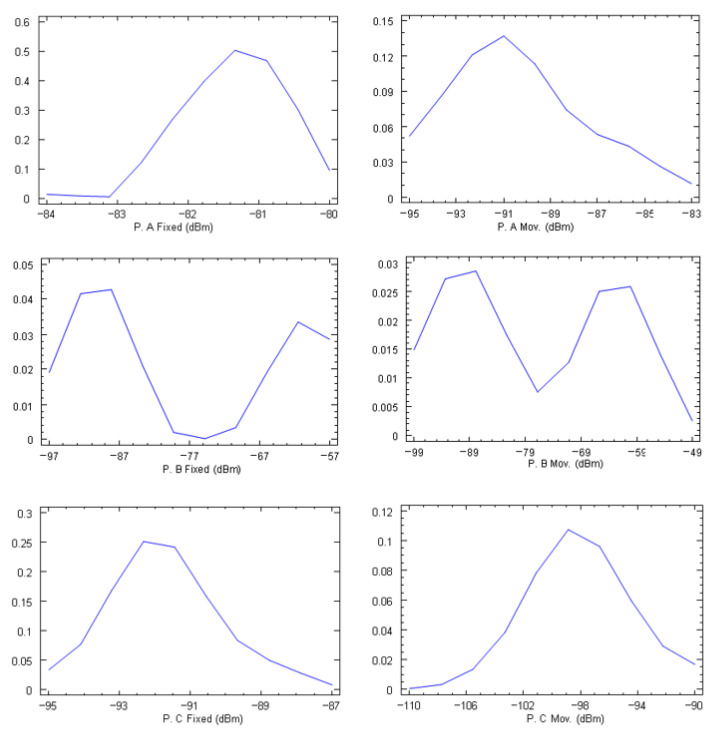
Density traces of the measurements taken at points A, B, and C.

**Table 1 sensors-24-03561-t001:** Comparison of measured and interpolated values.

Point #	LOS	EMF(V/m)	EMF dBV/m	Interp. (dBm)	Dif. (dB)
1	No	0.11	−19.17	−85.54	66.37
2	No	0.13	−17.72	−83.58	65.86
3	No	0.05	−26.02	−86.49	60.47
4	No	0.16	−15.92	−87.60	71.68
5	No	0.37	−8.64	−96.17	87.53
6	Yes	0.65	−3.74	−76.08	72.34
7	No	0.28	−11.06	−80.95	69.89
8	No	0.14	−17.08	−77.27	60.19
9	Yes	0.70	−3.10	−68.61	65.51
10	Yes	1.04	0.34	−65.49	65.83
11	No	0.10	−20.00	−94.24	74.24
12	Yes	1.47	3.35	−80.55	83.90
13	No	0.15	−16.48	−87.15	70.67
14	No	0.17	−15.39	−76.77	61.38
15	Yes	1.23	1.80	−66.76	68.56
16	Yes	0.72	−2.85	−77.77	74.92
17	No	0.31	−10.17	−73.47	63.30
18	Yes	1.00	0.00	−73.79	73.79
19	Yes	2.51	7.99	−66.04	74.04
20	No	0.36	−8.87	−71.02	62.14

**Table 2 sensors-24-03561-t002:** Comparison of averaged EMR300 and smartphone-measured values.

Point #	LOS	EMR300 (V/m)	EMR300 dBV/m	Mean (dBm)	Dif. (dB)
Point A Fixed	Yes	2.58	8.23	−81.32	89.55
Point A Movement	Yes	2.46	7.82	−82.76	90.58
Point B Fixed	Yes	5.40	14.64	−77.51	92.15
Point B Movement	Yes	5.16	14.25	−77.73	91.98
Point C Fixed	No	0.09	−20.92	−91.77	70.85
Point C Movement	No	0.09	−20.92	−97.94	77.02

## Data Availability

Dataset available on request from the authors.

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
