# Peer review of "Smartphone-Based Methodology Applied to Electromagnetic Field Exposure Assessment"

_sensors, 2024, doi:10.3390/s24113561_

Round 1

Reviewer 1 Report

Comments and Suggestions for Authors

Interesting paper. Good tool for fast EM field exposure estimation. I think the paper can be published.

Reviewer 2 Report

Comments and Suggestions for Authors

The work “Smartphone based methodology applied to Electromagnetic Field Exposure Assessment” is devoted to the description of the measurements of exposure to electromagnetic field by standard methods and using new approach (measurements of BTS signal strength by smartphone). Despite the fact that at the moment only initial verification work has been done, the results are of significant interest and can find application in the further development of the proposed method. The work can be published in the journal Sensors after responding to the comments below and making appropriate changes to the work:

1.        It is not very clear why only 20 fixed points were measured using a broadband meter, and 860 using a smartphone. It is clear that measurements using a smartphone are a faster approach that goes beyond the accepted standards of relevant measurements. However, the interval within which the measurements by broadband meter were carried out is quite wide. Perhaps it would have been worth measuring more points. Is this planned in the future? 

2.        2. A smartphone can also act as a source of EM fields. Was this taken into account in the work? How did the Bluetooth module work during the measurements? I mean the interference of the measured signal with harmonics of higher frequency signals. The authors should clarify this point in the article.

3.        The authors write (lines 205-207): “While in the case of the meter it is a measurement averaged over a 6‐minute interval at the same location, in the case of the smartphone it is a power measurement received by the terminal taken every 5 seconds along the indicated route.” Obviously, with short measurements using a smartphone, the measurement result also turns out to be very sensitive to short changes in the electromagnetic field (when new nearby sources of electromagnetic radiation appear, changes in the power of electromagnetic radiation from BTS or other sources at the current time, etc.), while more over long measurements this effect will be smoothed out. This should be noted and explained in the work.

4.        The authors write (lines 262-263): “For the 1.9 km2 area under study, the time used was 1 hour and 10 minutes compared to 4 hours for the traditional methodology.” From what is written in the work, it seems that measurements using a broadband meter took 4 hours for each of the 20 points studied. Could you please drown at this moment? If this is not so, then why were these measurements carried out over several days, and not within one day? Or it would be more correct to write about seconds in the case of measuring at one point using a smartphone and 4 hours in the case of measuring one point using a broadband meter.

5.        The authors should describe in more detail the further development of this work. Perhaps work is planned to increase the accuracy of comparison of the two methods used by increasing the number of points, stations, broadband meters, different smartphones, etc.

Comments on the Quality of English Language

I have no significant comments regarding the quality of English. I recommend additional proofreading of the work to improve individual formulations.

Author Response

Thank you very much for your suggestions and revisions to our proposal. We believe they are valuable and contribute to improving its quality. We have considered these in our revised version of the proposal. Please find our answers following your comments.

The work “Smartphone based methodology applied to Electromagnetic Field Exposure Assessment” is devoted to the description of the measurements of exposure to electromagnetic field by standard methods and using new approach (measurements of BTS signal strength by smartphone). Despite the fact that at the moment only initial verification work has been done, the results are of significant interest and can find application in the further development of the proposed method. The work can be published in the journal Sensors after responding to the comments below and making appropriate changes to the work:

  1. It is not very clear why only 20 fixed points were measured using a broadband meter, and 860 using a smartphone. It is clear that measurements using a smartphone are a faster approach that goes beyond the accepted standards of relevant measurements. However, the interval within which the measurements by broadband meter were carried out is quite wide. Perhaps it would have been worth measuring more points. Is this planned in the future? 

Response: Thank you very much for your comment. The density of points required in fixed site surveys varies greatly depending on the type of surface to be characterised, mainly due to its orography or the existence of line of sight to the emitters or multipath propagation. This analysis is included in the references of the paper [2]. For the proposed area of 1.9 km2, the minimum density to obtain adequate interpolations is approximately 8-10 points/km2. Obviously, the higher the point density, the better the interpolation accuracy. This implies a greater effort in carrying out the measurements. In this case, the total of 20 points took more than 4 hours. Each of these measurement points was averaged over 6 minutes and the displacements between points (average distance of about 300 metres) must be taken into account. On the other hand, in the case of smartphone measurements, the aim is to check whether measurements taken instantaneously (without averaging) can replace, by taking a larger number of measurements close to each other, fixed sites. We have included in the revised article an analysis of three additional points with measurements compared between the Narda meter and two smartphones.

  1. 2. A smartphone can also act as a source of EM fields. Was this taken into account in the work? How did the Bluetooth module work during the measurements? I mean the interference of the measured signal with harmonics of higher frequency signals. The authors should clarify this point in the article.

Response: Good suggestion. In the process of receiving the signals by the Smartphone, results that do not correspond to BTS signals are filtered out, so that downlink signals from different sources are controlled. It is true that the broadband meter receives additional signals such as Bluetooh or WiFi. An additional clarification has been included at the end of section 2.2.

“This also implies that these signals, which can be acquired by the EMR300, are not included in the measurement taken with the smartphone, which can be a source of deviation between the values taken by both methods.”

  1. The authors write (lines 205-207): “While in the case of the meter it is a measurement averaged over a 6‐minute interval at the same location, in the case of the smartphone it is a power measurement received by the terminal taken every 5 seconds along the indicated route.” Obviously, with short measurements using a smartphone, the measurement result also turns out to be very sensitive to short changes in the electromagnetic field (when new nearby sources of electromagnetic radiation appear, changes in the power of electromagnetic radiation from BTS or other sources at the current time, etc.), while more over long measurements this effect will be smoothed out. This should be noted and explained in the work.

Response: Thank you for this valuable suggestion. This comment has been included in the revised version after the above-mentioned paragraph.

“Measurements taken with the smartphone are subject to greater variability (fading or loss of signal) and may therefore lead to problems in the collected data. A further statistical analysis in section 3 discusses the statistical validity of measurements taken along a trajectory instead of in a fixed point.”

  1. The authors write (lines 262-263): “For the 1.9 km2 area under study, the time used was 1 hour and 10 minutes compared to 4 hours for the traditional methodology.” From what is written in the work, it seems that measurements using a broadband meter took 4 hours for each of the 20 points studied. Could you please drown at this moment? If this is not so, then why were these measurements carried out over several days, and not within one day? Or it would be more correct to write about seconds in the case of measuring at one point using a smartphone and 4 hours in the case of measuring one point using a broadband meter.

Response: The four hours indicated correspond to the time taken to carry out the measurement and additionally the journeys between the different fixed points. We have rewritten this sentence to clarify the correct meaning.

“For the 1,9 km2 area under study, the time taken to complete the route with the bicycle was 1 hour and 10 minutes. In the case of the fixed measurement sites, each of them was averaged for 6 minutes. We must add the time corresponding to the travel between points, so the total time spent in this method was 4 hours. This is an advantage because of the significant reduction in the time.”

  1. The authors should describe in more detail the further development of this work. Perhaps work is planned to increase the accuracy of comparison of the two methods used by increasing the number of points, stations, broadband meters, different smartphones, etc.

Response: Thank you for your suggestion. A paragraph on possible future work has been included in the conclusions section.

“In this proposal we have presented an alternative for the rapid characterization of an area with respect to cell phone emissions. It is not intended to replace the measurement by means of a personal exposure meter with a smartphone. For this, an extension of the research work on a much more exhaustive characterization of the terminal would be necessary to establish a relationship between the exposure values and the instantaneous values. The complete characterization of the gain of the cell phone is also indispensable to determine this relationship. Similarly, the influence on the exposure of services not covered in this work (FM, DTV, WiFi...) must be previously assessed when comparing the two values.”

Reviewer 3 Report

Comments and Suggestions for Authors

The paper mentions that there is an unknown offset factor between the data measured by smartphones and the data obtained by traditional methods. To improve the accuracy of the study, the authors may consider conducting a more in-depth calibration study to determine the specific conversion relationship between smartphone measurements and traditional measurements.

More comparative experiments can be conducted with professional electromagnetic field measurement equipment to determine the measurement deviation of smart phones under different environmental conditions and try to establish an accurate conversion model.

Adding in-depth statistical analysis to the dataset, for example, to consider changes in EMF exposure over different time periods, different weather conditions, or different user groups, can help provide a more complete understanding of the dynamic nature of EMF exposure.

Comments on the Quality of English Language

Although most of this paper is well written, there are still a few language problems in the manuscript.

Author Response

Thank you very much for your suggestions and revisions to our proposal. We believe they are valuable and contribute to improving its quality. We have considered these in our revised version of the proposal. Please find our answers following.

Comments:

The paper mentions that there is an unknown offset factor between the data measured by smartphones and the data obtained by traditional methods. To improve the accuracy of the study, the authors may consider conducting a more in-depth calibration study to determine the specific conversion relationship between smartphone measurements and traditional measurements.

More comparative experiments can be conducted with professional electromagnetic field measurement equipment to determine the measurement deviation of smart phones under different environmental conditions and try to establish an accurate conversion model.

Adding in-depth statistical analysis to the dataset, for example, to consider changes in EMF exposure over different time periods, different weather conditions, or different user groups, can help provide a more complete understanding of the dynamic nature of EMF exposure.

Response:

Thank you very much for the suggestion. We have performed additional experiments to evaluate possible relationships between fixed measurements and measurements taken with the smartphone. For this purpose, we have included points A, B and C in the study. In them we compared the six-minute averaging with taking measurements also for six minutes on the power meter tripod and making small random walks around the meter.

We have evaluated the statistical distributions (density traces) for the points taken with the smartphone when measured from a fixed point (in a manner equivalent to the broadband meter) and when measured by walking around the fixed-point set.

In both cases, the density traces follow the same behaviour, which allows us to state that taking multiple measurements from a single point (and then averaging or interpolating them by kriging) is statistically equivalent. The measurements obtained show a direct, though not equivalent, relationship between the broadband meter and smartphone values.

Due to the multiple factors involved (other wireless services present, reception of several frequencies simultaneously, different directions of arrival, different sensitivities of the meter and the smartphone, temporary signal fades...) it has not been possible to determine a single ratio value. Perhaps with further studies this factor can be determined (this has been indicated in the conclusions) and the usefulness of the proposal as a quick method to obtain rapid information on exposure has been emphasized.

Round 2

Reviewer 2 Report

Comments and Suggestions for Authors

The work “Smartphone based methodology applied to Electromagnetic Field Exposure Assessment” is devoted to the description of the measurements of exposure to electromagnetic field by standard methods and using new approach (measurements of BTS signal strength by smartphone). The authors gave comprehensive answers to the comments and made the necessary adjustments to the text. The work can be published in the journal Sensors without additional changes.